# Analytical Data Review to Determine the Factors Impacting Risk of Diabetes in North Al-Batinah Region, Oman

**DOI:** 10.3390/ijerph18105323

**Published:** 2021-05-17

**Authors:** Jabar H. Yousif, Firdouse R. Khan, Kashif Zia, Nahad Al Saadi

**Affiliations:** 1Faculty of Computing & Information Technology, Sohar University, P.O. Box 44, Sohar PCI 311, Oman; acaapub@gmail.com; 2Faculty of Business, Sohar University, P.O. Box 44, Sohar PCI 311, Oman; fkhan@su.edu.om

**Keywords:** chronic diseases, type 2 diabetes mellitus, multilinear regression, diabetes in Oman

## Abstract

Diabetes is one of the most widespread diseases resulting in an increase in mortality rate, and negatively affecting Oman’s economy. In 2019, an estimated 1.5 million deaths were directly caused by diabetes world health organization (WHO). The total number of diabetes cases among Omanis aged between 20 and 79 in 2015 is about 128,769, which increased in 2020 to 149,195. However, the total forecast number of diabetes cases will double in 2050 to 352,156. The healthcare spend on diabetes is 16.6%, which has triggered the need for the study. This research aims to review and analyze the prevailing situation around diabetes in Oman and its risk factors using multilinear regression tests, ANOVA, and descriptive analysis. Two hundred and fourteen samples were collected through a well-defined questionnaire using the purposive sampling technique. The study’s empirical results reveal that females, who were 79% of the respondents, have at least one of their family members as a diabetes patient; 41% of the respondents were affected by Type-2 (T2-DM) diabetes, and 38% were unaware of their type of diabetes. While, 91% of them were undergoing medication. The study indicates the risk factors that impact the diabetes number and type in North Oman, which include Aage with a rate of (0.595), followed by family history (0.560), smoking habits (0.530), and being overweight (0.435). Age plays an important role in the type of diabetes of the patients, and the types of diabetes impacts medication type. The type of diabetes has influenced the frequency of diabetes patients’ self-testing at home. Overall, 92% were rushed to the hospital or took medication immediately in case of a considerable variation in the readings; and 68% of the respondents complained that visiting the hospital is a costly affair and the health service is deteriorating.

## 1. Introduction

Improved health care facilities are important for the sustainable economic development of a country. Due to rapid changes in global lifestyle, governments and health care organizations must be on their toes all the time, able to gather real-time, on-demand information, and respond reactively. Towards this, the willingness and ability to adopt the latest technologies is key.

A chronic disease, also known as Non-Communicable Disease (NCDs), has been defined by the U.S. National Center for Health Statistics as a disease that lasts for three months or more [1]. It was reported that, in 2018, more people were admitted to hospitals in Oman for NCDs [2], particularly for heart diseases, hypertension (high blood pressure), stroke, cancer, chronic respiratory diseases, diabetes, arthritis, and asthma [3]. These diseases require sincere medical attention because they cause death and disability across the globe [4,5]. According to WHO report estimates that losses in gross domestic product (GDP) GDP worldwide [6], including both the direct and indirect costs of diabetes from 2011 to 2030, will reach a total of US 1.7 trillion, comprising both high-income countries with US 900 billion and low-income countries with US 800 billion. In 2019, an estimated 1.5 million deaths were directly caused by diabetes [7].

The total number of diabetes cases among Omanis aged between 20 and 79 years in 2015 is 128,769, which increased in 2020 to 149,195. However, the total forecast number of diabetes cases is expected to double in 2050 to reach 352,156, as shown in Figure 1.

Though diabetic clinics function in PHCs, complications due to T2 DM need to be primarily addressed [8]. High blood pressure, high blood glucose, obesity, and physical inactivity are commonly related factors [9]. However, the focus of this study is restricted to the situation of diabetes in the Sultanate of Oman. The commonly known types of diabetes are Type 1 diabetes (T1 DM)—insulin-dependent, Type 2 diabetes (T2 DM)—non-insulin-dependent, and Gestational Diabetes Miletus (GDM) [10]. It was believed that chronic diseases, particularly T2 DM, construed a major workload for clinicians in Omani hospitals in the early nineties. Most outpatient attendance, admissions, and bed occupancy in regional hospitals were caused by diabetes [11]. Moreover, it is noted that 71.9% of the total Omani population had at least one of the other chronic diseases and 12.61% are have diabetes [12].

As diabetes leads to life-threatening complications, diabetes patients are usually advised of healthy lifestyle habits, such as exercising and diet control. The normal practice here in Oman is that these patients visit the diabetes clinics at the Public Health Centers (PHCs) as outpatients, daily or once in two days. In Oman, every PHC has an exclusive diabetes clinic. The Health care staff of this diabetes clinic do general checkups—height/weight measurements, blood pressure, and glucose level, and provide medications for three months. However, patients are advised to keep track of their weight, blood pressure, and glucose levels daily. It is preferred that records are made of the health parameters at home and share them with the healthcare providers. However, during their visit to the clinic, most of them do not carry their recordings for references causing increased expenses in health care services.

Even though such follow up measures and preventive measures are being taken, it was reported that health care expenses have grown in Oman based on the ministry of health (MOH-Oman) reports from 148.9 million to 699.53 million from 2003 to 2015. In 2018, 16.6% of the healthcare budget was spent on diabetes, which has triggered the study’s need.

### Research Objectives

The aim of this paper is as follows:To review the prevailing situation of diabetes in the North Al-Batinah Region, Sultanate of Oman, andTo analyze the risk factors of diabetes in the country using multilinear regression tests, ANOVA, and descriptive analysis.

## 2. Review of Literature

Diabetes Miletus (DM) is considered a global cause of disability and a decisive risk factor for other diseases [13]. Alzaman stated that overweight, obesity, and Body Mass Index (BMI) increases the burden of diabetes in the Arab world [14]. There is a moderate risk of developing T2 DM among Omani adults within the next ten years if no preventive measures were taken [15]. There is a dire need for preventive measures from the future pandemic of diabetes in Oman as many people remain undiagnosed and others live without treatment [16]. Epidemiological changeover of diseases—CVDs, obesity, and diabetes in Oman are due to changes in the lifestyle of Omani society [17].

DM and obesity are more common in urban areas than in rural areas, and most of them are unaware that they have diabetes [18,19,20]. Most of the diabetes patients in Oman were of T2 DM type and not knowing physical activities, diet, blood glucose monitoring, etc., and there is a need for self-care education [21]. Self-maintenance and management of stable blood sugar levels among T2 DM patients are low with the education level and the higher the self-management of diabetes [22].

Due to deteriorating health services, people prefer private hospitals to public hospitals in Oman [23]. Hemoglobin A1c, Blood sugar (Random and fasting) levels for most diabetes patients seemed to be less within the safe range [24,25]. Diabetes leads to foot diseases, such as foot ulceration, infection, deformity, and lower limb amputation resulting in an increase in mortality rates in Oman [26]. A large number of diabetes patients in Oman have developed visual disabilities [27]. Diabetic retinopathy prevails mostly among those who have crossed 50 years [28]. There is a significant increase in the number of Omanis registered with DM and diabetic retinopathy in Oman in the past two decades [29]. Most of the Omani families have at least one of the parents with diabetes and having the risk of developing diabetes with at least one complication—high blood pressure, coronary artery disease, or retinopathy [30,31].

After thoroughly going through the above review of literature, the factors such as lifestyle—smoking habit, overweight, physical activity, family history, blood sugar level, family history of diabetes, knowledge of diabetes, self-care, medication were identified and taken into consideration for the research study. A theoretical framework is illustrated in Figure 2.

## 3. Material and Methods

A well-defined questionnaire was prepared to carry out the research study, including the factors mentioned above, then distributed to the selected participants. The participants in this questionnaire did not report their written acceptance regarding their participation in the research, but received the questionnaire by text message and voluntarily answered it. The collected data were filtered using a purposive sampling technique. Out of the 332 samples collected from the population, 214 samples were diabetes patients only considered for the research study [32]. Further, the data were compiled, tabulated, and tested for robustness before proceeding with other tests, such as multilinear regression tests (gender-wise), ANOVA, cross-tabulation relational analyses, descriptive analysis, etc. The cross-tabulation test between (Age, Type of diabetes, Medication type, the types of diabetes) and frequency of testing at the home of diabetes of the patients.

From the review of the literature, it can be summarized that the type of diabetes is an important factor, which influences the medication type of diabetes patients. It also influences the frequency of testing at home. Blood glucose status is also an important factor, which affects the frequency of testing at home.

Blood glucose status is a factor that influences the frequency of visiting diabetes clinic and the tests prescribed by doctors is an important factor, which affects the frequency of visiting diabetes Clinic.

Thus, the following hypotheses are drawn:

**Hypotheses** **1a.***Age plays an important role in the type of diabetes of the patients*.

**Hypotheses** **1b.***The types of diabetes play an important role in the medication type of diabetes patients*.

**Hypotheses** **1c.***The types of diabetes play an important role in the frequency of testing at home*.

**Hypotheses** **2a.***The Blood Glucose Status plays an important role in the frequency of testing at home*.

**Hypotheses** **2b.***Testing Blood Sugar at home play an important role in the monitoring of blood sugar per day*.

**Hypotheses** **3a.***The kind of tests prescribed by Doctors play an important role in the frequency of visiting diabetes clinic*.

**Hypotheses** **3b.***Blood Glucose Status plays an important role in the frequency of visiting a diabetes clinic*.

## 4. Results and Discussion

Table 1 shows the demographic details of the respondents. Age is an important factor, which influences the type of diabetes of the patients. It is claimed that age plays an important role in the type of diabetes of the patients.

Table 2 shows the cross-tabulation between the age and the type of diabetes, the medication type and the types of diabetes and frequency of testing at home and the types of diabetes of the patients.

It is evident from Table 2 that the *p* values are less than 0.05. Hence there is a relationship between the age of the patients and the type of diabetes and also a relationship between types of diabetes and the medication type undertaken. Therefore, Hypothesis 1a–c, i.e., age plays an important role in the type of diabetes; the types of diabetes play an important role in the medication type of diabetes patients, and the types of diabetes play an important role in the frequency of testing at home are proved.

It is evident from Table 3 that the *p*-value is less than 0.05. Hence, there is a relationship between blood glucose status and the frequency of testing at home. There is a relationship between blood glucose status and the frequency of visiting diabetes clinics. Therefore, Hypotheses 2a,b, that the blood glucose status plays an important role in the frequency of testing at home and the testing blood sugar at home plays an important role in monitoring blood sugar per day is proved.

It is evident from Table 4 that the *p*-value is less than 0.05 for the cross-tabulation between the kind of test prescribed and the frequency visiting diabetes clinic, whereas the *p*-value > 0.05 for the cross-tabulation between High/Low Blood sugar status. Hence there is a relationship between the kind of tests prescribed by Doctors and the frequency of visiting diabetes clinics, whereas there is NO relationship between blood glucose status and the frequency of visiting diabetes clinics.

In other words, the kind of tests prescribed by Doctors plays an important role in the frequency of visiting diabetes clinics, whereas the blood glucose status plays an important role in the frequency of visiting diabetes clinics is disproved.

## 5. Multicollinearity Test Results

The results of the multicollinearity tests are presented in Table 5 as follows: 

Table 5 shows that the tolerance value of all independent variables has a value >0.1 and has a Variance Inflation Factor value (VIF) <10. Therefore, it can be concluded that there is no multicollinearity between the independent variables in the regression model. Therefore, the obtained results of the multilinear regression tests are as shown in Table 6, Table 7, Table 8 and Table 9.

From Table 9, it can be seen that the *p*-value for all the dependent variables (Sig.) is less than 0.05, which clearly shows that the types of diabetes are dependent on the independent variables viz. the gender, age, smoking habit, overweight, and family history. Thus, the obtained linear regression model can be written as in Equation (1):TD = −1.617 + 0.304 G + 0.595 A + 0.530 S + 0.295 OW + 0.560 FH(1)
where TD—Types of Diabetes, G—Gender, A—Age, S—Smoking habit, OW—Overweight and FH—Family History.

Based on observing the coefficients of the independent variables, it can be said that age has the highest impact on the types of diabetes (0.595), followed by Family History (0.560), Smoking habit (0.530), Gender (0.304), and Overweight (0.295). Filtering from the population, and selecting only the female samples, the multilinear regression test is carried out again, and the results are as shown in Table 10, Table 11, Table 12 and Table 13.

From the above table (Table 13), it can be seen that the *p*-value for all the dependent variables (sig.) is less than 0.05, which clearly shows that the types of diabetes are dependent on the independent variables viz. Age, Overweight, and Family History. Thus, the obtained linear regression model can be written as in Equation (2).
TD = −0.042 + 0.464 A + 0.435 OW + 0.437 FH(2)
where TD—Types of Diabetes, A—Age, OW—overweight and FH—Family History.

Based on observing the coefficients of the independent variables, we can say that age has the highest impact on the Types of diabetes (0.464), followed by Family History (0.437), and Overweight (0.435).

## 6. Discussion

Table 1 shows that most of the respondents (79%) are females, and it was also found that 70% of them were obese (Overweight), and most of them (58%) responded that at least one of their family members was a diabetes patient. Even though 80% of them did not have a Smoking habit, they reported diabetes. While, 41% of the respondents were affected by Type 2 (T2 DM) diabetes, and 38% were unaware of the type of diabetes. It was also found that 91% of them were undergoing medication of different types; 70% of them visiting the doctors continually and cared for their diabetes regularly; and 83% of the respondents reported that their blood glucose status varied often. Most of them (77%) suggested that they were prescribed to undergo general tests by the doctors during their visits; 94% said that they measured their blood glucose status daily; 78% reported that they measured their blood glucose status on their own at home; 88% of the respondents said that they kept a record of self-measured readings at home; and 59% reported recording their status in papers, while the rest reported maintaining records either in hospital files or in mobile phones.

The comparison with other studies should undertake under the same conditions. Therefore, it is difficult to compare our results with other studies implemented in different environments. However, the literature review indicates some of the risk factors that impact the diabetes type and prevalence. These risk factors, include overweight and obesity [12], family history, blood sugar level, and age of those who have crossed 50 years [28]. These factors were identified and taken into consideration for the research study. The results of this study are in line with other studies that examine the risk factors, including age as the main factor followed by Family History, Smoking habit, and Overweight.

## 7. Conclusions

It is observed that age plays an important role in the type of diabetes of patients, and the types of diabetes impacts the medication type of diabetes patients. The type of diabetes also influenced the frequency of diabetes patients self-testing at home. Many patients rushed to the hospital or took medication immediately when they indicated considerable variations in the readings. One-third of the respondents worried and reported that they do not have any kind of access to their records and readings. While, 81% opined that using the related application in the mobile phones was helpful to save and retrieve their readings to communicate the same easily, and 66% reported that they were already using such applications through their mobile phones and tabs.

The primary investigation was to determine the risk factors that impact the types of diabetes and their speared rate. This study observed the following risk factors:Based on Equation (1), it can say that age has the highest impact on the types of diabetes (0.595), followed by Family History (0.560), Smoking habit (0.530), Gender (0.304), and Overweight (0.295).Based on Equation (2), it can say that age has the highest impact on the types of diabetes (0.464), followed by Family History (0.437) and Overweight (0.435).From the literature review, we can summarize that diabetes type is an important factor influencing the medication type of diabetes patients. It also affects the frequency of testing at home. This study also observed that age plays an essential role in the type of diabetes of patients and the types of diabetes impact the medication type of diabetes patients.

The study’s limitations are the low number of participants, which impacts the generalization of all Oman results. Besides, a shortage of reliable data related to the exact number of patients with diabetes because we obtained different numbers based on WHO and MOH-Oman. This impact limits our analysis scope, which it considers a notable obstacle in finding a significant relationship.

From the above, as an implication of the study, the following suggestions were made:The government should take initiatives to run awareness campaigns on Obesity and diabetes.A database of diabetes patients need to be created so that regular advice, follow-up be made in an easy and smooth manner.Facilities for their regular check up on their blood glucose sugar level can be made available, either by special camps or by providing them with self-measuring apparatuses.Age-wise classification of diabetes patients should be made and special attention to be made to the age-old diabetes patients, and women in particular.Records of such patients can be linked through online facilities so that they can upload their self-measured readings to retrieve them anytime, even during emergency situations.Facilities for obtaining timely appointments can be made at every level starting from the roots—public health centers to the apex level—hospitals.At the regional/Wilayat level, special camps can be established to curb and educate using increasing diabetes awareness campaigns, to be held on a periodic/regular interval.

## Figures and Tables

**Figure 1 ijerph-18-05323-f001:**
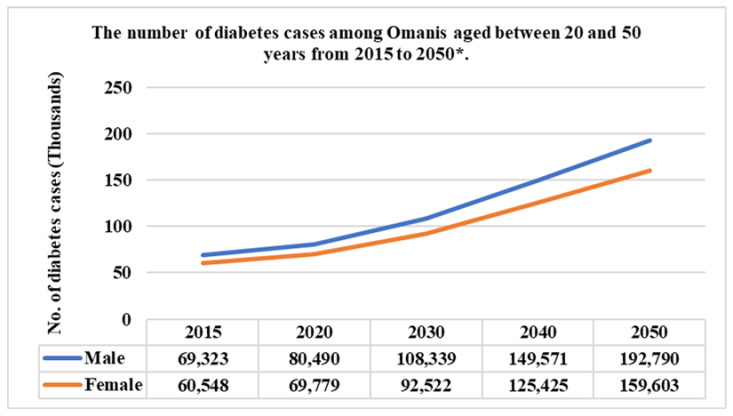
Diabetic Patients in Oman [8], * is the predicted numbers.

**Figure 2 ijerph-18-05323-f002:**
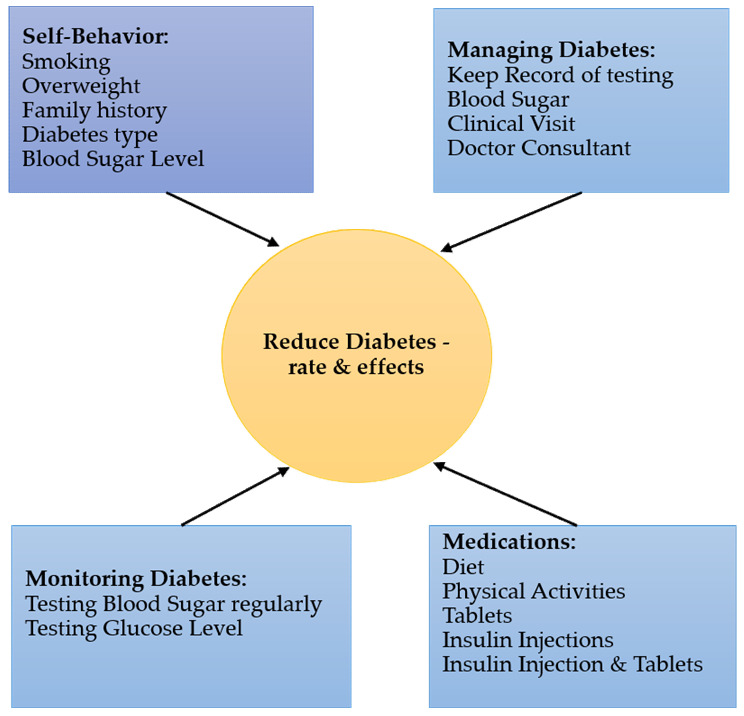
Theoretical Framework.

**Table 1 ijerph-18-05323-t001:** Demographic details of the respondents.

Characteristics		Frequency	%
Nationality	Omani	214	100.0
	Expatriate	0	0.0
Gender	Male	45	21.0
	Female	169	79.0
Age	Less than 20 years	15	7.0
	20–<40 years	62	29.0
	40–<70 years	118	55.1
	70 years and above	19	8.9
Smoking habit	Never	171	79.9
	Sometimes	25	11.7
	Always	18	8.4
Overweight	Yes	122	57.0
	No	64	29.9
	Don’t know	28	13.1
Anyone in the family has diabetes	Yes	124	57.9
	No	15	7.0
	Don’t know	75	35.1
Type of diabetes you have	Type 1 (T1 DM)	31	14.5
	Type 2 (T2 DM)	88	41.1
	Gestational (GDM)	17	8.0
	Don’t know	78	36.4
Type of Medication you take	Tablets	131	61.2
	Insulin Injection	17	8.0
	Tablets and Insulin Injection	3	1.4
	Diet/Sports like life style changes only	43	20.1
	Nothing	20	9.3
Visiting the Diabetes Clinic	1–4 times a month	124	57.9
	More than 4 times a month	28	13.1
	1–4 times a year	62	29.0
How often you had low/high blood sugar (Blood Glucose status)	Never	37	17.3
	Sometimes	49	22.9
	Always	128	59.8
Kind of tests doctor suggests you	General	166	77.6
	Specific	8	3.7
	Both	40	18.7
The doctor advised you on the readings you measured at your home	Never	105	49.1
	Sometimes	66	30.8
	Always	43	20.1
Monitoring blood sugar at least once a day	Very important	131	61.2
	Important	70	32.7
	No Idea	13	6.1
Testing your blood sugar at home	Never	47	22.0
	Sometimes	117	54.6
	Always	50	23.4
Keeping a record of your self-measured readings	Never	39	18.2
	Sometimes	137	64.1
	Always	38	17.7
Recording your readings	In Papers	127	59.3
	In Hospital file	74	34.6
	In Mobile phone	13	6.1
Your reaction to low or high blood sugar	Take medication at home	27	12.6
	Go to the hospital	159	74.3
	Either of the above	20	9.4
	Do not react	8	3.7
You have access to your results	Never	80	37.4
	Sometimes	103	48.11
	Always	31	4.5
Getting hospital appointments is a costly affair	Yes	146	68.2
	No	37	17.3
	Do not know	31	14.5
Having a smartphone/tablet	Yes (I use applications)	141	65.9
	Yes (use for calling only)	60	28.0
	No	13	6.1
Using mobile application beneficial to save and give feedback to diabetic patients	Yes	174	81.3
	No	11	5.1
	Do not know	29	13.6

Source: Questionnaire.

**Table 2 ijerph-18-05323-t002:** Cross tabulation combination (i).

	Types of Diabetes						
	T1 DM	T2 DM	GDM	Do Not Know	Total	χ^2^	*p*-Value
**Age**							
<20	2	3	3	7	15	65.318	0.000
20–40	20	14	13	15	62		
41–70	6	66	1	45	118		
>70	3	5	0	11	19		
Total	31	88	17	78	214		
**Medication Type**							
Only life style changes	10	14	4	15	43	59.158	0.000
Tablets	11	66	5	49	131		
Insulin Injection	9	4	1	3	17		
Tablets & Insulin	1	1	0	1	3		
Nothing	0	3	7	10	20		
Total	31	88	17	78	214		
**Frequency of testing at home**							
Never	9	12	3	23	47	13.511	0.036
Sometimes	16	47	9	45	117		
Always	6	29	5	10	50		
Total	31	88	17	78	214		

**Table 3 ijerph-18-05323-t003:** Cross tabulation combination (ii).

	Frequency of Testing at Home					
	Never	Sometimes	Always	Total	χ^2^	*p*-Value
**Blood Glucose Status**						
Never	17	17	3	37	21.351	0.000
Sometimes	5	26	18	49		
Always	25	74	29	128		
Total	47	117	50	214		
**Monitoring blood sugar at least once a day**						
Very Important	19	74	38	131	24.835	0.000
Important	19	40	11	70		
No idea	9	3	1	13		
Total	47	117	50	214		

**Table 4 ijerph-18-05323-t004:** Cross tabulation Combination (iii).

	Frequency of Visiting Diabetes Clinic					
	1–4 Times a Month	More than 4 Times in a Month	1–4 Times a Year	Total	χ^2^	*p*-Value
**Kind of Tests Prescribed**						
General Tests	105	21	40	166	12.119	0.016
Specific Tests	5	1	2	8		
Both	14	6	20	40		
Total	124	28	62	214		
**High/Low Blood Sugar Status**						
Never	18	4	15	37	4.507	0.342
Sometimes	77	15	36	128		
Always	29	9	11	49		
Total	124	28	62	214		

**Table 5 ijerph-18-05323-t005:** Multicollinearity Test Results.

Model	Tolerance	VIF
Gender	0.754	1.326
Age	0.930	1.076
Smoking habit	0.749	1.335
Overweight	0.502	1.994
Family history	0.517	1.933

**Table 6 ijerph-18-05323-t006:** Multilinear Regression Test Results/Variables Entered/Removed.

Model	Variables Entered	Variables Removed	Method
1	Age, Gender, Smoking habit, Overweight, Family history		Enter

Note: Dependent Variable: Types of Diabetes; All requested Variables entered.

**Table 7 ijerph-18-05323-t007:** Multilinear Regression Test Results/Model Summary.

Model	R	R Square	Adjusted R Square	Std. Error of the Estimate
1	0.747	0.557	0.547	0.752

Note: Predictors: (constant), Age, Gender, Smoking habit, Overweight, Family history.

**Table 8 ijerph-18-05323-t008:** Multilinear Regression Test Results/ANOVA ^a^.

Model	Sum of Squares	df	Mean Square	F	Sig.
Regression	148.118	5	29.624	52.370	0.000 ^b^
Residual	117.658	208	0.566		
Total	265.776	213			

^a^ Dependent Variable: Types of Diabetes; ^b^ Predictors: (constant), Age, Gender, Smoking habit, Overweight, Family history.

**Table 9 ijerph-18-05323-t009:** Multilinear Regression Test Results/Coefficients.

Model	Unstandardized Coefficients		Standardized Coefficients	t	Sig.
	B	Std. Error	Beta		
(constant)	−1.617	0.417		−3.873	0.000
Gender	0.304	0.145	0.111	2.090	0.038
Age	0.595	0.068	0.421	8.801	0.000
Smoking habit	0.530	0.097	0.290	5.438	0.000
Overweight	0.295	0.102	0.189	2.900	0.004
Family history	0.560	0.076	0.471	7.340	0.000

Note: Dependent Variable: Types of Diabetes.

**Table 10 ijerph-18-05323-t010:** Multilinear Regression results of female samples/Variables Entered/Removed.

Model	Variables Entered	Variables Removed	Method
1	Age, Overweight, Family History	…	Enter

Note: Dependent Variable: Types of Diabetes; All requested Variables entered.

**Table 11 ijerph-18-05323-t011:** Multilinear Regression results of female samples/Model Summary.

Model	R	R Square	Adjusted R Square	Std. Error of the Estimate
1	0.672	0.451	0.441	0.819

Note: Predictors: (constant), Age, Overweight, Family History.

**Table 12 ijerph-18-05323-t012:** Multilinear Regression results of female samples/ANOVA ^a^.

Model	Sum of Squares	df	Mean Square	F	Sig.
Regression	91.068	3	30.356	45.244	0.000 ^b^
Residual	110.707	165	0.671		
Total	201.775	168			

^a^ Dependent Variable: Types of Diabetes; ^b^ Predictors: (constant), Age, Overweight, Family History.

**Table 13 ijerph-18-05323-t013:** Multilinear Regression results of female samples/Coefficients.

Model	Unstandardized Coefficients		Standardized Coefficients	t	Sig.
	B	Std. Error	Beta		
(constant)	−0.042	0.281		−0.149	0.882
Age	0.464	0.083	0.328	5.615	0.000
Overweight	0.435	0.154	0.278	2.824	0.005
Family history	0.437	0.117	0.371	3.748	0.000

Note: Dependent Variable: Types of diabetes.

## Data Availability

Online resource. DOI:10.17632/bvdb3nbtxs.1 (accessed on 16 May 2021).

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
