# Peer review of "Analytical Data Review to Determine the Factors Impacting Risk of Diabetes in North Al-Batinah Region, Oman"

_ijerph, 2021, doi:10.3390/ijerph18105323_

Round 1

Reviewer 1 Report

The authors present a title of the manuscript “Analytical Data Review to determine the factors Impacting Risk of Diabetes in Sultanate of Oman” and an objective that is The aim of this paper is to review the pre-vailing situation of diabetes in the Sultanate of Oman and to analyze the risk factors of diabetes in the country.

However, in the results and discussion section, the authors do not mention the risk factors of diabetes in the country

Authors should mention the statistical analysis used in the methodology section. Research Methodology & Hypotheses. To carry out the research study, a well-defined questionnaire was prepared including the above-mentioned factors and the same was distributed. The data thus collected was  filtered using a purposive sampling technique. Out of the 332 samples collected from the opulation, 214 samples - diabetes patients only considered for the research study  (DOI:10.17632/bvdb3nbtxs.1). Further, the data was compiled, tabulated and tested for  robustness, before proceeding with other tests to arrive at the conclusion.

Author Response

We want to take this opportunity to thank the editorial office and the reviewers for their valuable comments and informative feedback, which enrich the article.  Please view the attached file for a point-by-point rebuttal.

Reviewer 2 Report

The article entitled "Analytical Data Review to determine the factors Impacting 2Risk of Diabetes in Sultanate of Oman" written by Yousif et al., is found significant and interesting. Yousif et al., conducted the study to understand the risk factors responsible for diabetes in Omani Population and presented the analytical approach to understand these risk factors.

However, this article needs minor corrections before considering for publication in International Journal of Environmental Research and Public Health.

  • In abstract section data presented seems old (2015), Provide the latest update of diabetes patient (2019 or 2020) in Oman published by WHO or Ministry of health in Oman.
  • Sample size taken to represent the Sultanate of Oman diabetic population is very low, it should be increased to get the correct analytical data or to understand the risk factors. Otherwise title should be changed to particular states or province where the study was conducted.
  • English language and spellings should be corrected throughout the manuscript, in addition at some places rephrasing required.
  • Keywords presented should follow journal instruction.
  • Line no 30 to 40, reference provided in text should following journal guideline. It should be presented serial wise.
  • In introduction section, lot of information regarding the non-communicable diseases (NCDs) is provided which is not required; it could be reduce so reader can focus over the objective of the study rather than general information.
  • Presentation of article should be made clear, especially in introduction and review section, keeping in minds of the reader.
  • Figure 1 shows the total no. of diabetic patient in 2014 and 2016 is 83785 and 88898 However in abstract the data shows of around 6669 as per Ministry of Health Oman. This could create discrepancies of the report.
  • Objective of the study should be clearly provided
  • Literature review section can be added into introduction section; however author should avoid too much general information.
  • Material method should include the ethical statement as well as the approval granted from authorities.
  • Presented theoretical hypothesis is well defined by several author concerning as the risk factor, What takes author to choose such factor which is explained earlier in several reports.
  • Going through the conclusion of this study, it seems bit long, reduce it or see the journal guidelines.

Author Response

(The authors gave the same response as above.)

Reviewer 3 Report

Thank you for the opportunity to review your paper about the analytical data review to determine the factors impacting risk of Diabetes in Sultanate of Oman. Congratulations on the paper presented and the study described. Below is some feedback intended to help you strengthen the manuscript.

Abstract

The abstract includes the main information about the study. However, I suggest that the authors add the main conclusions of the study and implications of the results found.

Introduction

In my opinion, the introduction is well constructed and has the necessary information to understand the problem under study. I suggest that the authors add the objectives of the study at the end of this introduction.

Review of Literature

I suggest changing the title of this chapter to Background. I also suggest that the authors address public health responses for the population with Diabetes. How are health services organized and what kind of response do they give?

Research Methodology & Hypotheses

I suggest changing the title of this chapter to Material and Methods.

Methodological options should be described. The authors do not identify the type of study, the type of sampling and what are the criteria for selecting the sample, the description of the data collection method, the ethical procedures and how the data analysis was carried out.

Results & Discussion

I suggest that the authors separate the presentation of the results and the discussion. The results found are not discussed or compared with other studies. Authors should discuss the results they have found and compare them with other studies in the field.

At the end of the discussion, the authors should develop the implications of the results of this study and the main limitations or difficulties encountered. Suggestions for improvement should also be presented at the end of the discussion.

Conclusions

In the conclusions, the authors must summarize the main results found (the authors do not need to present relative frequency values).

Author Response

(The authors gave the same response as above.)

Round 2

Reviewer 1 Report

I have no observations on the re-submmited manuscript

Reviewer 3 Report

Dear authors. Thank you for making the changes I suggested. I think the article has become clearer and more complete. Just a reminder that as you have decided to separate the results of the discussion, you should correct the chapter name "results and discussion". I wish many future successes.